# Nonlinear MCMC for Bayesian Machine Learning

**James Vuckovic**
james@jamesvuckovic.com

## Abstract

We explore the application of a nonlinear MCMC technique first introduced in [1] to problems in Bayesian machine learning. We provide a convergence guarantee in total variation that uses novel results for long-time convergence and large-particle ("propagation of chaos") convergence. We apply this nonlinear MCMC technique to sampling problems including a Bayesian neural network on CIFAR10.

## 1  Introduction

Characterizing uncertainty is a fundamental problem in machine learning. It is often desirable for a machine learning model to provide a prediction *and* a measure of how "certain" the model is about that prediction. Having access to a robust measure of uncertainty becomes particularly important in real-world, high risk scenarios such as self-driving cars [2–4], medical diagnosis [5, 6], and classifying harmful text [7].

However, despite the need for uncertainty in machine learning predictions, it is well known that traditional ML training, i.e. based on optimizing an objective function, frequently does not provide robust uncertainty measures [8], yielding overconfident predictions for popular neural networks such as ResNets [9]. [1] An appealing alternative to the traditional optimization paradigm for ML is the Bayesian probabilistic framework, due to its relatively simple formulation and extensive theoretical grounding; see for example [10].

From the probabilistic perspective of machine learning [10], one combines a prior $P(\theta)$ over the parameter space $\theta \in \Theta$ and a likelihood of the data given model parameters $P(\mathcal{D}|\theta)$ using Bayes' rule to obtain a posterior over the parameters $P(\theta|\mathcal{D}) \propto P(\mathcal{D}|\theta)P(\theta)$. The "traditional" approach in machine learning is to *optimize* the posterior (or the likelihood) to obtain $\theta^* \in \arg\max P(\theta|\mathcal{D})$ and generate predictions via $P(y|x, \mathcal{D}) = P(y|x, \theta^*)$. However, if we adopt the Bayesian approach, the posterior characterizes the uncertainty about the parameters of the model (i.e. epistemic uncertainty), which can propagate to uncertainty about a prediction by *integration*: $P(y|x, \mathcal{D}) = \int P(y|x, \theta)P(\theta|\mathcal{D})\mathrm{d}\theta$. This paper studies the problem of how to approximate this integration with samples from $P(\theta|\mathcal{D})$.

### 1.1  Contributions

- Our main contribution is the novel analysis of a modification of the general nonlinear Markov Chain Monte Carlo (MCMC) sampling method from [1] to obtain quantitative convergence guarantees in both the number of iterations and the number of samples.

- We apply the general results from above to determine the convergence of two specific nonlinear MCMC samplers.

- In experiments, we compare these nonlinear MCMC samplers to their linear counterparts, and find that nonlinear MCMC provides additional flexibility in designing sampling algorithms with as good, or better, performance as the linear variety.

---

[1] In Appendix C.2.4, we provide an experiment that demonstrates this effect.

36th Conference on Neural Information Processing Systems (NeurIPS 2022).

## 1.2 Background

**Bayesian ML & MCMC.** In Bayesian machine learning, the "computationally difficult" step is integration since the integral $\int P(y|x,\theta)P(\mathrm{d}\theta|\mathcal{D})$ is not analytically solvable except for rare cases. In practice, one typically uses a Monte Carlo approximation such as

$$\int P(y|x,\theta)P(\theta|\mathcal{D})\mathrm{d}\theta \approx \frac{1}{N}\sum_{i=1}^{N}P(y|x,\theta^i), \quad \text{where} \quad \theta^i \overset{iid}{\sim} P(\theta|\mathcal{D})$$

where the expected error in this approximation is well known to converge to zero like $\mathcal{O}(1/\sqrt{N})$ by the Central Limit Theorem (CLT). There are various approaches to sampling from $P(\theta|\mathcal{D})$, but Markov chain Monte Carlo (MCMC) is perhaps the most widely used. The basic idea of MCMC is to use a Markov transition kernel $\mathcal{T}$ with stationary measure $P(\theta|\mathcal{D})$ to simulate a Markov chain $\theta_{n+1} \sim \mathcal{T}(\theta_n, \bullet)$ that converges rapidly to $P(\theta|\mathcal{D})$. In this case, we can estimate

$$\int P(y|x,\theta)P(\theta|\mathcal{D})\mathrm{d}\theta \approx \frac{1}{N}\sum_{i=1}^{N}P(y|x,\theta_\infty^i) \approx \frac{1}{N}\sum_{i=1}^{N}P(y|x,\theta_{n_{\mathrm{sim}}}^i)$$

where $n_{\mathrm{sim}}$ is some large number of simulation steps and $\{\theta_n^i\}_{n=0}^{\infty}$ are independent Markov chains governed by $\mathcal{T}$.

The basic problem is then to design an efficient transition kernel $\mathcal{T}$. There is a vast body of literature studying various choices of $\mathcal{T}$; some well known choices are the Metropolis-Hastings algorithm [11, 12], the Gibbs sampler [13], the Langevin algorithm [14–17], Metropolis-Adjusted Langevin [18, 19], and Hamiltonian Monte Carlo [20–24].

However, there are various challenges in Bayesian ML that make applying these samplers difficult in practice. One challenge is that the posterior $P(\theta|\mathcal{D})$ can be highly multimodal [25, 26], which makes it difficult to ensure that the Markov chain $\theta_n$ explores all modes of the target distribution. One can combat this issue by employing auxiliary samplers that explore a more "tractable" variation of $P(\theta|\mathcal{D})$ [27, 28]. Other methods that empirically improve posterior sampling quality include tempering [29–32], RMSProp-style preconditioning [33], or adaptive MCMC algorithms [34, 35] such as the popular No U-Turn Sampler [36].

**Nonlinear MCMC.** Another class of powerful MCMC algorithms, which is less-studied in the context of Bayesian ML, arises from allowing the transition kernel $\mathcal{T}$ to depend on the distribution of the Markov chain as in $\theta_{n+1} \sim \mathcal{T}_{\mathrm{Distribution}(\theta_n)}(\theta_n, \bullet)$. This approach gives rise to so-called *nonlinear* MCMC since $\{\theta_n\}$ is no longer a true Markov chain. Nonlinear Markov theory is a rich area of research [37–42] and has strong connections to nonlinear filtering problems [43, 44], sequential Monte Carlo [45, 46], and nonlinear Feynman-Kac models [47]. One can replace $\mathrm{Distribution}(\theta_n)$, which is often intractable, with an empirical estimate $\mathrm{Distribution}(\theta_n) \approx \frac{1}{N}\sum_{i=1}^{N}\delta_{\theta_n^i}$ to obtain *interacting* particle MCMC (iMCMC, or iPMCMC) methods; see for example [48, 49, 1, 50, 51].

Our view is that nonlinear MCMC offers some appealing features that traditional linear MCMC lacks. One such feature is the ability to leverage *global information* about the state space $\Theta$ contained in $\mathrm{Distribution}(\theta_n^i)$ to improve exploration, a central issue in Bayesian ML. Another feature is the increased flexibility of nonlinear MCMC algorithms, which can be leveraged to correct biases that are introduced by other design decisions in MCMC for Bayesian ML such as tempering. These features will be explored empirically in Section 4.

However, the theoretical analysis of nonlinear Markov MCMC presents an added difficulty in that the particles $\{\theta_n^1, \ldots, \theta_n^N\}$ of an interacting particle system are now *statistically dependent*. This means that, in addition to studying the long-time behaviour which is classical in MCMC [52, 53], one must study the large-particle behaviour separately to obtain Monte Carlo estimates since the CLT does not apply. One such large-particle behaviour is the propagation of chaos [41], which is the tendency for groups of interacting particles to become independent as the number of particles, $N$, increases; see [41]. We will need both of these elements — long-time convergence and propagation of chaos — to properly characterize the convergence of nonlinear MCMC.

**Other Sampling Methods.** Finally, let us mention that MCMC is certainly not the only way to obtain Monte Carlo sample estimates in Bayesian ML; some popular examples include MC dropout [8], black-box variational inference [54], and normalizing flows [55, 56].

## 1.3 Common Notation

Let $\mathcal{P}(\mathbb{R}^d)$ be the set of probability measures on the measurable space $(\mathbb{R}^d, \mathscr{B}(\mathbb{R}^d))$. For $\mu \in \mathcal{P}(\mathbb{R}^d)$ and $f \in \mathcal{B}_b(\mathbb{R}^d) := \{f : \mathbb{R}^d \to \mathbb{R} \mid f \text{ is bounded}\}$, we will denote $\mu(f) := \int f \mathrm{d}\mu$. If $K : \mathbb{R}^d \times \mathscr{B}(\mathbb{R}^d) \to [0, 1]$ is a Markov kernel[2] then we will denote $Kf(x) := \int f(y)K(x, \mathrm{d}y)$ and $\mu K(\mathrm{d}y) := \int \mu(\mathrm{d}x)K(x, \mathrm{d}y)$. Finally, for $\overline{y} := \{y^1, \dots, y^N\} \subset \mathbb{R}^d$, we will denote the empirical measure of $\overline{y}$ as $m(\overline{y}) := \frac{1}{N} \sum_{i=1}^{N} \delta_{y^i} \in \mathcal{P}(\mathbb{R}^d)$.

# 2 Nonlinear MCMC

In this section, we outline the family of MCMC algorithms that will be studied in rest of the work. We will use general notation for simplicity but this difference is merely cosmetic; the "target distribution" $\pi$ in this section corresponds directly to $P(\theta|\mathcal{D})$ from the previous section.

## 2.1 Nonlinear Jump Interaction Markov Kernels

To specify a MCMC algorithm, we must specify the Markov transition kernel. The family of nonlinear Markov kernels that we will be studying was introduced in [1] and is a mixture of a linear kernel, denoted $K$, and a nonlinear jump-interaction kernel indexed by a probability measure $\eta$, denoted $J_\eta$, to obtain

$$K_\eta(x, \mathrm{d}y) := (1 - \varepsilon)K(x, \mathrm{d}y) + \varepsilon J_\eta(x, \mathrm{d}y) \tag{1}$$

where $\varepsilon \in ]0, 1[$ is the mixture hyperparameter. The Markov kernel $K_\eta$ will be the main object of interest throughout this paper. We will give specific examples of $J_\eta$ in Section 2.2, which were also introduced in [1]. Despite building on the constructions of [1], this work proceeds in some different substantially directions; see Appendix A.1 for more details.

**Mean Field System.** Now we show how the kernel $K_\eta$ can be used to construct a Markov chain. Following [1], we use an auxiliary Markov chain $\{Y_n\}$ with transition kernel $Q$ on the same state space as $K_\eta$ (i.e. $\mathbb{R}^d$) to obtain the nonlinear Markov chain $\{(Y_n, X_n)\}_{n=0}^{\infty}$ defined by

$$\begin{cases} Y_{n+1} \sim Q(Y_n, \bullet) \\ \eta_{n+1} := \text{Distribution}(Y_{n+1}) \quad Y_0 \sim \eta_0, \ X_0 \sim \mu_0 \\ X_{n+1} \sim K_{\eta_{n+1}}(X_n, \bullet) \end{cases} \tag{2}$$

where $\mu_0, \eta_0 \in \mathcal{P}(\mathbb{R}^d)$ are the initial distributions and $\sim$ denotes "sample from". One should interpret this as a sequence of steps where first we sample the auxiliary state $Y_{n+1}$ from $Q$, then we obtain the distribution of $Y_{n+1}$ denoted $\eta_{n+1}$, and we use this distribution to index the primary kernel $K_{\eta_{n+1}}$ and obtain a sample $X_{n+1}$. We sample $X_{n+1}$ with probability $(1 - \varepsilon)$ from the linear kernel $K$, and with probability $\varepsilon$ it will "jump" according to $J_{\eta_{n+1}}(X_n, \bullet)$. Because the Markov dynamics depend on $\text{Distribution}(Y_n)$, we call this a "mean field system".

**Interacting Particle System.** One issue with the mean field system (2) is the fact that computing $\text{Distribution}(Y_{n+1})$ is generally impossible except in special cases. Hence, to get a viable simulation algorithm, we must approximate $\text{Distribution}(Y_n)$, and we do this by replacing $\text{Distribution}(Y_n)$ with its empirical measure estimated from a set of $N$ particles $\overline{Y}_n := \{Y_n^1, \dots, Y_n^N\}$ as follows:

$$\begin{cases} Y_{n+1}^i \sim Q(Y_n^i, \bullet) \\ \eta_{n+1}^N := m(\overline{Y}_{n+1}) \quad Y_0^i \overset{iid}{\sim} \eta_0, \ X_0^i \overset{iid}{\sim} \mu_0, \ i = 1, \dots, N. \\ X_{n+1}^i \sim K_{\eta_{n+1}^N}(X_n^i, \bullet) \end{cases} \tag{3}$$

## 2.2 Application to MCMC

Now we detail how to apply $K_\eta$ and the Markov chains (2) and (3) to MCMC. In particular, we must understand how to choose $Q, K, J_\eta$ such that $K_\eta$ will be invariant w.r.t. a target distribution $\pi$.

---

[2] i.e. $K(x, \bullet)$ is a probability measure $\forall x \in \mathbb{R}^d$ and $K(\bullet, A)$ is measurable $\forall A \in \mathscr{B}(\mathbb{R}^d)$

As is usually the case in probabilistic inference problems, we will assume that the target distribution $\pi$ is known only up to a normalizing constant and that it has a density, also denoted $\pi$. We also make the simplifying assumption that $Q$ has an invariant measure $\eta^\star$ (also with density denoted $\eta^\star$) i.e. $\eta^\star Q = \eta^\star$. This is not burdensome; in practice we can, and will, obtain $Q$ from a *linear* MCMC algorithm for some choice of $\eta^\star$. In fact, being able to choose $\eta^\star$ is a powerful design parameter of our methods as we will see in Section 4. We will also assume that the linear kernel $K$ is $\pi$-invariant, i.e. $\pi K = \pi$.

To see how we can ensure that $\pi$ is $K_\eta$-invariant, consider the fact that we will design $Q$ s.t. $\eta_n := \mathrm{Distribution}(Y_n)$ converges to $\eta^\star$. This means we will eventually be sampling from the kernel $K_{\eta^\star}$ and we already have $\pi$-invariance of $K$. Therefore, if we arrange for $J_{\eta^\star}$ to be $\pi$-invariant, $\pi$ will be invariant for $K_{\eta^\star}$ since

$$\pi K_{\eta^\star} = (1-\varepsilon)\pi K + \varepsilon \pi J_{\eta^\star} = (1-\varepsilon)\pi + \varepsilon \pi = \pi.$$

Intuitively, if the auxiliary chain converges to a steady state and the jumps in that steady state preserve $\pi$ (and $K$ preserves $\pi$), then so will $K_{\eta^\star}$. Now the remaining task is to design nonlinear interaction kernels $J_\eta$ that will yield good performance; we detail two choices below.

**Boltzmann-Gibbs Interaction.**    The first choice of $J_\eta$ we will investigate, from [1], relies on the Boltzmann-Gibbs transformation [47], which we now explain. Let $G : \mathbb{R}^d \to ]0, \infty[$ be a potential function; then the Boltzmann-Gibbs (BG) transformation is a nonlinear mapping $\Psi_G : \mathcal{P}(\mathbb{R}^d) \to \mathcal{P}(\mathbb{R}^d)$ defined by

$$\Psi_G(\mu)(\mathrm{d}x) := \frac{G(x)}{\mu(G)}\mu(\mathrm{d}x) \quad \text{or equivalently} \quad \int f(x)\Psi_G(\mu)(\mathrm{d}x) := \int f(x)\frac{G(x)}{\mu(G)}\mu(\mathrm{d}x)$$

for any $f \in \mathcal{B}_b(\mathbb{R}^d)$ and whenever $\mu(G) \neq 0$. This transformation has many interesting properties and been extensively studied in [47] and related works.

To use the BG transformation in MCMC, we will assume that the densities $\pi$ and $\eta^\star$ are positive[3] and make the choice that $G(x)$ will be the function

$$G(x) := \frac{\pi(x)}{\eta^\star(x)}. \tag{4}$$

With this choice, we get an interaction kernel $J_\eta^{BG}(x, \mathrm{d}y) := \Psi_G(\eta)(\mathrm{d}y)$. We can easily see that

$$\eta^\star(G) = \int \frac{\pi}{\eta^\star}\mathrm{d}\eta^\star = \int \mathrm{d}\pi = 1 \quad \text{and} \quad \Psi_G(\eta^\star)(\mathrm{d}x) = \frac{G(x)}{\eta^\star(G)}\eta^\star(\mathrm{d}x) = \frac{\pi(x)}{\eta^\star(x)}\eta^\star(\mathrm{d}x) = \pi(\mathrm{d}x),$$

i.e. $\Psi_G$ is the multiplicative "change of measure" from $\eta^\star$ to $\pi$. Hence the first nonlinear Markov kernel we will investigate is

$$K_\eta^{BG}(x, \mathrm{d}y) := (1-\varepsilon)K(x, \mathrm{d}y) + \varepsilon\Psi_G(\eta)(\mathrm{d}y). \tag{5}$$

From the remarks above, is clear that $\pi$ is $K_{\eta^\star}$-invariant.

**Accept-Reject Interaction.**    The second choice of jump interaction we will study, also introduced in [1], is a type of accept-reject interaction related to the Metropolis-Hastings algorithm. For the *same* choice of potential function $G$ in (4), we can define the acceptance ratio

$$\alpha(x, y) := 1 \wedge \frac{G(y)}{G(x)} = 1 \wedge \frac{\pi(y)\eta^\star(x)}{\eta^\star(y)\pi(x)} \quad \text{and the quantity} \quad A_\eta(x) := \int \alpha(x, y)\eta(\mathrm{d}y)$$

for $\eta \in \mathcal{P}(\mathbb{R}^d)$. Hence we can define the accept-reject interaction kernel as[4]

$$J_\eta^{AR}(x, \mathrm{d}y) := \alpha(x, y)\eta(\mathrm{d}y) + (1 - A_\eta(x))\delta_x(\mathrm{d}y).$$

We can interpret this jump interaction as: starting in state $x$, we jump to a new state distributed according to $\eta(\mathrm{d}y)$ with probability $\alpha(x, y)$ (i.e. accept the proposed jump) and remain the in current state with probability $1 - A_\eta(x)$ (i.e. reject the proposed jump). This is a form of "adaptive Metropolis-Hastings" in which the proposal distribution evolves over time as the distribution of the auxiliary Markov chain. Hence we obtain the accept-reject nonlinear jump interaction kernel

$$K_\eta^{AR}(x, \mathrm{d}y) := (1-\varepsilon)K(x, \mathrm{d}y) + \varepsilon[\alpha(x, y)\eta(\mathrm{d}y) + (1 - A_\eta(x))\delta_x(\mathrm{d}y)]. \tag{6}$$

We note that $\pi$ is also $J_{\eta^\star}^{AR}$-invariant; see Proposition 3 in Appendix G for a simple calculation.

---

[3] this can be relaxed to $\pi \ll \mu$ and $\mu \ll \pi$

[4] Given $f \in \mathcal{B}_b(\mathbb{R}^d)$, we can also write this as $J_\eta^{AR}f(x) = \int[f(y) - f(x)]\alpha(x, y)\eta(\mathrm{d}y) + f(x)$

**Simulation.** Let us note briefly that using both $K_\eta^{BG}$ and $K_\eta^{AR}$ in (3) produce interacting particle systems that can, and will, be simulated. The simulation is relatively straightforward, see Appendix A for pseudocode implementing the nonlinear MCMC algorithms we have now constructed.

## 3 Convergence Analysis

We will now study whether the nonlinear MCMC algorithms based on $K_\eta$ from Section 2 — i.e., the interacting particle system (3) with the restrictions on $K, Q, J_\eta$ from Section 2.2 — will actually converge to the target distribution $\pi$. In other words, we would like to estimate $\|\mu_n^N - \pi\|$ for some suitable notion of distance on $\mathcal{P}(\mathbb{R}^d)$, where $\mu_n^N := \mathrm{Distribution}(X_n^1)$ is the distribution of a single particle (it doesn't matter which particle as the $X_n^i$ are *exchangeable*).

The nonlinear nature of $K_\eta$ makes this analysis more difficult than of a linear MCMC method. We break the problem into two parts: one studying the convergence of the mean-field system (2) as the number of steps $n \to \infty$, and one studying the convergence of the interacting particle system (3) to the mean field system as the number of particles $N \to \infty$. This will allow us to apply the triangle inequality as follows:

$$\|\mu_n^N - \pi\| \leq \underbrace{\|\mu_n^N - \mu_n\|}_{\text{large-particle convergence}} + \underbrace{\|\mu_n - \pi\|}_{\text{long-time convergence}}$$

where $\mu_n := \mathrm{Distribution}(X_n)$ is the distribution of the mean-field system. Crucially, our analysis of the large-particle limit is *uniform* in the number of steps $n$, which will allow us to establish bounds above that hold as $n \to \infty$. The actual result is contained in Theorem 1.

While our analysis does not rely on heavy mathematical machinery, to state the full set of conditions and results for long-time and large-particle convergence — each of which is a substantial result in its own right — would occupy too much space in the main text. Instead, we will state the main result in Theorem 1, which is essentially a corollary of the long-time and large-particle analyses Theorems 2 and 3 in Appendices E and F respectively, and below we will sketch the general arguments used in those appendices. The proofs of the main results are in Appendix F.2. Note that our analysis, and the results we obtain, are novel and not found in [1]; see Appendix D for an elaboration.

The following result is stated in terms of the *total variation* metric, defined here for $\mu, \nu \in \mathcal{P}(\mathbb{R}^d)$ as $\|\mu - \nu\|_{tv} := \sup_{\|f\|_\infty \leq 1} |\mu(f) - \nu(f)|$ where $\| \bullet \|_\infty$ is the sup-norm on $\mathcal{B}_b(\mathbb{R}^d)$.

**Theorem 1.** *[Convergence of Nonlinear MCMC] Under suitable conditions on $K_\eta$ and $Q$, there exist fixed constants $C_1, C_2, C_3 > 0$, a function $\mathcal{R} : [0, \infty[ \to [1, \infty[$, and $\rho > 0$ s.t.*

$$\|\mu_n^N - \pi\|_{tv} \leq C_1 \frac{1}{N} \mathcal{R}(1/N) + C_2 \rho^n + C_3 n \rho^n.$$

♦

Let us make a couple of remarks:

- This result shows that, to control the approximation error $\|\mu_n^N - \pi\|_{tv}$, it does not necessarily suffice to run the MCMC algorithm for a large number of steps $n$, since if $n \to \infty$ but $N < \infty$ then our bound on $\|\mu_n^N - \pi\| \not\to 0$. However, this approximation cannot lead to arbitrarily bad results: Theorem 1 provides a quantitative upper bound on how much the MCMC algorithm can be biased. This behaviour is supported empirically; in Figure 5 of Appendix C.1.4 we provide a clear illustration of how changing $N$ significantly affects the bias of our nonlinear MCMC methods while having no effect on the bias of linear MCMC, as expected.

- This result uses total variation, which is a strong metric that represents a worst-case over *all* bounded functions $f$ (up to rescaling by $\|f\|_\infty$). It is entirely possible that, for many choices of practical $f$, the approximation will be better as we will see empirically.

- The constant $\rho$ is, roughly speaking, the slower of the rate of convergence for $Q$ and for $K$. Hence if $K, Q$ are chosen to be efficient samplers with fast convergence, this will result in $\rho \ll 1$ and hence $\mu_n^N$ will also converge quickly.

- In our specific samplers $K_\eta^{BG}$ and $K^{AR}$, we will see in Appendix G that $\mathcal{R}$ is a monotonically increasing function that is lower-bounded by 1. Hence, as $N \to \infty$, $\frac{1}{N}\mathcal{R}(\frac{1}{N}) \to 0$ as expected.

A corollary of Theorem 1 is that that we regain a Monte Carlo estimate for the interacting particle system. This result is essentially due to [41] Theorem 2.2.

**Corollary 1.** *[Adapted from [41], Theorem 2.2] Suppose that Theorem 1 applies to $K_\eta$. Let $\overline{X}_n := \{X_n^1, \ldots, X_n^N\}$ be the interacting particle system from (3). Then for every $n \in \mathbb{N}$ and $f \in \mathcal{B}_b(\mathbb{R}^d)$ we have*

$$\lim_{N\to\infty} \mathbb{E}\left[ \left\| \frac{1}{N}\sum_{i=1}^N f(X_n^i) - \mu_n(f) \right\| \right] = 0.$$

♦

This corollary directly relates to the application of Bayesian ML we are interested in, where we would have $f(\theta) = P(y|x, \theta)$.

## 3.1 Long-Time Bounds

There are two main ingredients in the general result on long-time convergence: ergodicity of $K$ and $Q$, and Lipschitz regularity of the interaction kernel $\eta \mapsto J_\eta$. These, along with other technical conditions, produce an estimate of the form $\|\mu_n - \pi\| \leq C_2\rho^n + C_3 n\rho^n$ where $\| \bullet \|$ is a weighted total variation norm. The full statement is in Theorem 2 of Appendix E.

**Ergodicity of $K$ and $Q$.** A fundamental requirement of our results is that the *linear* building blocks of $K_\eta$ must converge to their respective stationary measures in an appropriate metric. This type of result is now standard in the Markov chain literature, and we use a result from [57] for $K$ and a result from [1] for $Q$. The former is actually able to ensure that $K$ is a contraction on $\mathcal{P}(\mathbb{R}^d)$ w.r.t. a suitable weighted total variation; we use this feature repeatedly in our analysis.

**Lipschitz Regularity of $J_\eta$.** We also need that $\eta \mapsto J_\eta$ is Lipschitz-continuous w.r.t. a weighted total variation norm on Markov kernels (the Lipschitz constant does not have to be $< 1$). This regularity is used to translate the convergence of $\eta_n \to \eta^\star$ (as guaranteed by the ergodicity of $Q$) into convergence of $J_\eta \to J_{\eta^\star}$ with the Lipschitz estimate $\|J_{\eta_n} - J_{\eta^\star}\| \lesssim \|\eta_n - \eta^\star\|$. In Appendix G, we verify this analytically for $J_\eta^{BG}$ and $J_\eta^{AR}$, see Lemma 5 and [1] Proposition 5.3.

## 3.2 Large-Particle Bounds

To study the large-particle behaviour, we would like to measure how close subset of $q \in \{1, \ldots, N\}$ interacting particles $\{X_n^1, \ldots, X_n^q\} \subset \{X_n^1, \ldots, X_n^N\} =: \overline{X}_n$ from (3) is to being i.i.d. according to the mean-field measure $\mu_n$. This analysis was pioneered in [41] under the name "propagation of chaos" and formalizes the intuition that, as $N \to \infty$, the influence of any individual particle $\to 0$.

To state this more precisely, first note that if we had random variables $Z^i \overset{iid}{\sim} \eta \in \mathcal{P}(\mathbb{R}^d)$ then the joint distribution of $\overline{Z} := \{Z^1, \ldots, Z^N\}$ would be $\eta^{\otimes N}$. Hence, as $N \to \infty$ for the interacting particles $\overline{X}_n$ at time $n$, we expect the distribution of $\{X_n^1, \ldots, X_n^q\}$, denoted $\mu_n^{q,N}$, to get closer to the distribution of i.i.d. mean field particles from (2), denoted $\mu_n^{\otimes q}$. In other words, we expect $\|\mu_n^{q,N} - \mu_n^{\otimes q}\|_{tv} \to 0$ as $N \to \infty$. The full statement of this result is Theorem 3 of Appendix F.

The main condition in our propagation of chaos result is another type of regularity for $\eta \mapsto J_\eta$ which basically requires that, if one approximates a distribution $\eta \in \mathcal{P}(\mathbb{R}^d)$ by its empirical measure $m(\overline{Y})$ where $\overline{Y} := \{Y^1, \ldots, Y^N\}$ and $Y^i \overset{iid}{\sim} \eta$, then $J_{m(\overline{Y})} \to J_\eta$ as $N \to \infty$. More precisely, there should be a function $\mathcal{R} : [0, \infty[ \to [1, \infty[$, which is ideally nondecreasing, s.t.

$$|\mathbb{E}[J_{m(\overline{Y})}^{\otimes q} f(x)] - J_\eta^{\otimes q} f(x)| \lesssim \frac{q^2}{N}\mathcal{R}(q^2/N) \quad \forall x \in \mathbb{R}^d \text{ and } f \in \mathcal{B}_b(\mathbb{R}^d) \text{ with "oscillations" } \mathrm{osc}(f) \leq 1.$$

The expectation is taken over $\eta^{\otimes N}$, and "oscillations" are defined precisely in Appendix D. This inequality is essentially a total variation regularity since we can alternately write $\|\mu - \nu\|_{tv} = \sup\{|\mu(f) - \nu(f)| \mid f \in \mathcal{B}_b(\mathbb{R}^d), \; \mathrm{osc}(f) \leq 1\}$ [47].

### 3.3 Analysis of Specific Interaction Kernels

The main result Theorem 1 is in terms of conditions on a general $K_\eta$ (i.e. general choices of $K, Q, J_\eta$). To apply this result to the samplers in Section 2, we must establish if these conditions hold for $K_\eta^{BG}$ and $K_\eta^{AR}$. Fortunately this can be done analytically; in Appendix G, we present conditions under which the Lipschitz regularity (see Lemma 5 and [1] Proposition 5.3) and large-particle regularity (see Corollaries 3 and 5) hold. These results, particularly for $K_\eta^{BG}$, are interesting and rely on novel techniques for controlling the various quantities, sometimes improving over previous methods. Due to space constraints, the results for $K_\eta^{BG}$ and $K_\eta^{AR}$ are in Corollaries 2 and 4 from Appendix G.

## 4 Experiments

In this section, we detail two experiments designed to explore how one might apply the nonlinear MCMC methods developed in the previous sections to Bayesian machine learning.[5] Let us state explicitly that our aim is *not* to achieve state-of-the-art with these experiments, nor do we claim that this method will necessarily lead to state-of-the-art results on a particular task. Rather, the aims of these experiments are: to demonstrate that nonlinear MCMC can be applied successfully to large-scale problems; to compare linear vs nonlinear methods to understand what benefits and drawbacks nonlinear MCMC offers compared to linear MCMC in practice; and to develop some recipes for choosing the various hyperparameters and samplers that determine a nonlinear MCMC method.

### 4.1 Two-Dimensional Toy Experiments

First, we use a toy setting of two-dimensional distributions to compare the relative benefits of linear and nonlinear MCMC. A benefit of this simple setting is that the multimodal toy distributions can be exactly sampled. This gives us the opportunity to quantify the quality of our samples via an unbiased estimator of the Maximum Mean Discrepancy (MMD) metric on $\mathcal{P}(\mathbb{R}^d)$ [58]. This approach stands in contrast to many previous works, which use simplistic distributions (e.g. Gaussians) with analytically tractable statistics to measure quality. See Appendix C.1 for an overview of our methodology.

**Setup.** Our setup will compare the Metropolis Adjusted Langevin Algorithm (MALA) [18] (see Appendix B for an overview of MALA) with the nonlinear BG and AR samplers using MALA for the kernels $K, Q$ in our nonlinear setup from Section 2. This will allow us to examine the effects of the nonlinearity in $K_\eta^{AR}$ and $K_\eta^{BG}$ while controlling for the type of sampler used and its hyperparameters.

The main difference between the linear and nonlinear algorithms, aside from the interaction itself, is the extra "design knob" to control in the form of the choice of the auxiliary density $\eta^\star$. Below, we show how one can use $\eta^\star$ to incorporate additional insight to guide the sampling, such as regions of the state space to explore. In our experiments, we choose $\eta^\star$ to be a centered, 2-dimensional Gaussian with a large variance ($\Sigma = 4I_2$ for the circular MoG and two-rings densities, and $\Sigma = 20I_2$ for the grid MoG). This conveys "coarse-grained" information of roughly where the support of the target density is located – in this case, a neighbourhood of $(0, 0)$. See Table 2 in Appendix C.1 for a full account of our experimental settings.

**Results.** From Figure 1, we see that having a simple auxiliary density with good coverage of the support of the target distribution is quite helpful. In all three examples, one or both of the nonlinear samplers outperformed the equivalent linear sampler in the empirical MMD metric. For the two most challenging densities, the "two rings" and "grid MoG" distributions, the improved exploration is particularly evident. We also include an analysis of the runtime of the algorithms in Appendix C.1.3.

**Comparison With [1].** We also compared the performance of our methods with those of [1] in this two-dimensional toy setting. See Appendix C.1.4 for an overview of the results; they support all of the theoretical considerations and design principles we have introduced in this paper.

---

[5] The code used in our experiments can be found at https://github.com/jamesvuc/nonlinear-mcmc-paper. See also Appendix A for a discussion of the implementation details.

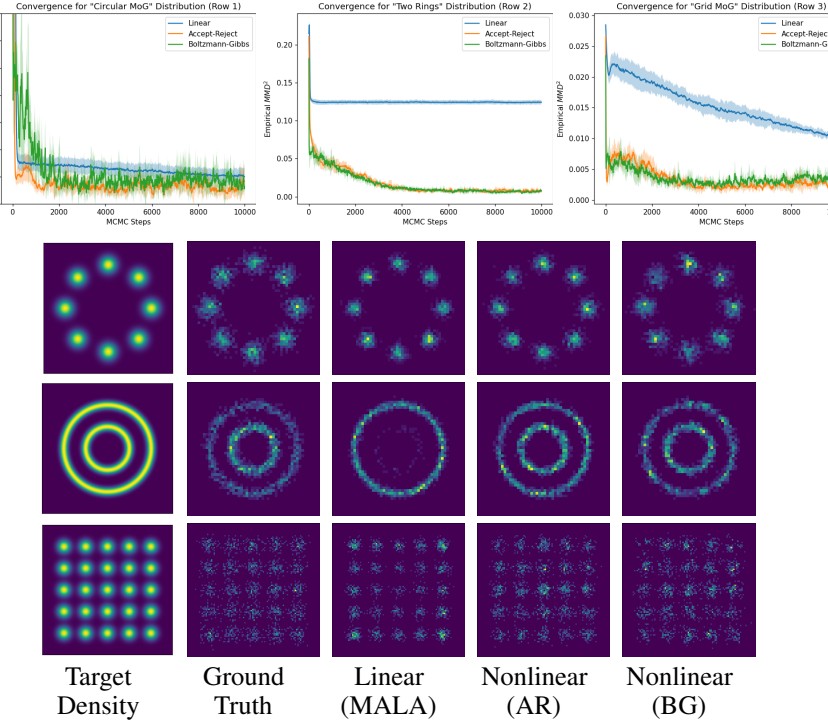

Figure 1: Visualizations of the 2d experiment. The top row shows the empirical MMD-squared plotted over number of sampled steps, where the shaded region is $\pm 1$ standard deviation with 5 independent runs. The bottom three rows show histograms for the $N = 2000$ samples of the Circular Mixture of Gaussians (MoG) density [59], the Two Rings density [59], and the Grid Mixture of Gaussians density [60] respectively.

## 4.2 CIFAR10

### 4.2.1 Setup

To examine the properties of the nonlinear sampler outside of a toy setting, we have also implemented a Bayesian neural network on the CIFAR10 dataset. We use a likelihood $P(y|x, \theta)$ parameterized by a ResNet-18 convolutional neural network [61] and a Gaussian prior $P(\theta)$ on the parameters of this neural network which are combined to form a posterior $P(\theta|\mathcal{D}_{\text{train}}) \propto P(\theta) \prod_{(x_i, y_i) \in \mathcal{D}_{\text{train}}} P(y_i|x_i, \theta)$. The goal is to sample $\theta^i$, $i = 1, \ldots, N$ from this posterior. See Table 4 in Appendix C.2 for a full account of the experimental settings.

To deal with the fact that this sampling problem is very high-dimensional ($d \approx 11M$) and $|\mathcal{D}_{\text{train}}|$ is large ($|\mathcal{D}_{\text{train}}| = 60,000$) we use a variety of techniques:

1. We sample minibatches $\widehat{\mathcal{D}}$ of size 256 to obtain the surrogate target density $P(\theta|\widehat{\mathcal{D}})$ [17].
2. We use an RMSProp-like "preconditioned" Langevin algorithm, called RMS-Langevin or RMS-ULA, as in [33] for the auxiliary sampler $Q$; see Appendix B for details on this sampler. As shown in [33], this sampler is biased.
3. We use tempering, wherein we aim to sample from $\pi$ or $\eta \propto P(\theta|\mathcal{D}_{\text{train}})^{1/\tau}$ where $\tau$ is a small number. This substantially improves mixing for hard-to-sample distributions such as $P(\theta|\mathcal{D}_{\text{train}})$ at the cost of bias since we are no longer sampling from the true posterior [32].

Using our nonlinear algorithm presents a novel opportunity to correct the bias introduced by tempering. For our experiments, we pick $\eta^\star \propto P(\theta|\mathcal{D}_{\text{train}})^{1/\tau}$ and $\pi = P(\theta|\mathcal{D}_{\text{train}})$. This means that the auxiliary chain $Y_n$ explores a tempered version of the target, whereas the target chain $X_n$ (in theory) explores the true target distribution. This is a novel strategy that is made possible by being able to select $\eta^\star$ almost independently of the target $\pi$. We study the case when $\pi$ is tempered as well.

For our experiments, we use the RMS-Langevin sampler as the baseline, and we also use it for the auxiliary sampler $Q$. For the target sampler, it is not possible to use the RMS-Langevin algorithm because the smoothed square-gradient estimate is incompatible with the discontinuities (i.e. jumps) introduced by the nonlinear interaction. Instead, for the linear sampler $K$ we use the unadjusted Langevin algorithm, ULA, [16] (see Appendix B). We investigate both test accuracy and calibration error [9] to assess performance.

Table 1: Results for CIFAR10 experiments. $\pm$ represents 1 standard deviation on 5 random seeds. The tempered results are using $\tau = 10^{-4}$. See Appendix C.2 for an overview of expected calibration error. We also compute the maximum calibration error in Appendix C.2. All Expected Calibration Error numbers are multiplied by $10^2$ in this table.

| | **Test Accuracy** ($\uparrow$) | | **Expected Calibration Error** ($\downarrow$) | |
|---|---|---|---|---|
| Algorithm | Non-Tempered | Tempered | Non-Tempered | Tempered |
| Linear | $85.01_{\pm 0.10}$ | $85.01_{\pm 0.19}$ | $0.24_{\pm 0.02}$ | $0.26_{\pm 0.014}$ |
| Nonlinear (BG) | $84.28_{\pm 0.28}$ | $84.74_{\pm 0.08}$ | $0.14_{\pm 0.03}$ | $0.16_{\pm 0.03}$ |
| Nonlinear (AR) | *Diverged* | $84.67_{\pm 0.23}$ | *Diverged* | $0.15_{\pm 0.05}$ |

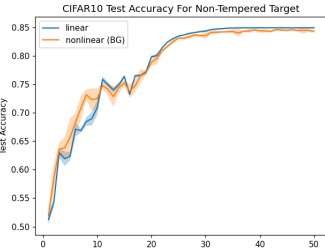 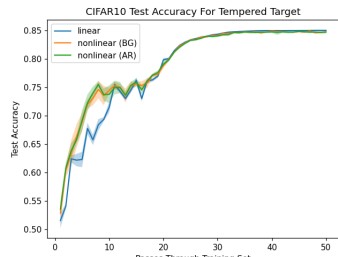

Figure 2: Evaluation of test accuracy during sampling for CIFAR10. The shaded areas represent $\pm 1$ standard deviation for 5 random seeds. For readability, we omit the AR interaction curve on the non-tempered result since it diverged and it distorts the scale of the plot. For completeness, all the curves for the non-tempered case are plotted in Appendix C.2.2, Figure 6.

### 4.2.2 Results

**Linear vs Nonlinear.** From Table 1, we see that the linear (RMS-Langevin) sampler has slightly higher, but comparable test accuracy to the nonlinear samplers. This is likely because RMS-Langevin algorithm has better stability around the regions of high probability due to its adaptive stepsize scaling. However, from Figure 2, we see that for both the tempered and non-tempered cases, the nonlinear interaction appears to benefit during early exploration. This is an expected and desired property of these nonlinear samplers, which incorporate global information about the sampler state (in this case, the relative potential $G$ of each auxiliary chain's state) and are able to emphasize those states with higher probability. However, the linear method is able to eventually explore the relevant regions of the state space, and the difference disappears. See Appendix C.2.5 for a comparison linear vs nonlinear performance scaled by the number of gradient evaluations.

**Tempering Vs Non-Tempering.** The linear MCMC sampler is always tempered in our experiments so there should not be any statistically significant difference in the linear case. For the nonlinear sampler, the tempered version has slightly higher accuracy; this trend is also observed in [32]. On the other hand, the calibration errors are the same for both tempered and non-tempered variants. This is somewhat surprising, given the aggressive tempering used, and one would expect that this reduces the variance of the posterior estimate.[6] This observation can perhaps be explained by the fact that we are using $N = 10$ samples which may not be enough to accurately change the tempered auxiliary distribution $\eta^\star$ into the non-tempered primary distribution $\pi$ in the jump interaction.

---

[6] As $\tau \to 0$, this $\theta \sim P(\theta|\mathcal{D}_{\text{train}})^{1/\tau}$ converges to the maximum *a posteriori* estimate with zero variance.

**Calibration.** Considering the expected calibration error (ECE) [9], in Table 1 we see that the nonlinear method has statistically significantly lower ECE ($p \ll 0.05$) compared to the linear method. We hypothesize that this is due to a tension between the RMS scaling of the gradient which improves the efficiency of each MCMC step but at the cost of bias, which may be measurable in the form of calibration error. The Langevin algorithm is also biased, but is generally known to have good convergence properties [16] and much less is known about the RMS-Langevin variant. By using our nonlinear setup, we are able to aggressively explore the auxiliary distribution without sacrificing calibration on the target distribution.

## 5 Conclusion

In this paper, we have studied the theoretical and empirical properties of nonlinear MCMC methods. We have obtained powerful theoretical results to characterize the convergence of our MCMC methods, and we have applied these methods to Bayesian neural networks. The results on BNNs are comparable to, but not better than, the linear methods we studied. We hypothesize that this is because more investigation into choosing the best auxiliary density $\eta^\star$ is required; our choice is simplistic and may not be optimal. This hypothesis is supported by our toy experiments, which show significant improvement when $\eta^\star$ is able to incorporate some additional insight into the problem. How to do this in high dimensions is an exciting direction for future research.

**Broader Impact.** There are benefits and drawbacks to the nonlinear MCMC methods we describe. The benefits are mainly that properly accounting for uncertainty in machine learning will lead to better real-world outcomes for high-value scenarios such as self-driving cars or medical imaging. The drawbacks are that MCMC methods require $\mathcal{O}(N)$ storage and computations relative to the $\mathcal{O}(1)$ for deterministic methods; in fact, our nonlinear method would require $2N$ resources compared with $N$ for a linear method (see also Appendices C.1.3 and C.2.5). If our algorithms were applied to a large swath of ML "as is", this would mean a substantial increase in the energy consumption required for experimentation and deployment, worsening an already substantial issue in the field.

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
