# OpenReview forum: "Nonlinear MCMC for Bayesian Machine Learning"
_NeurIPS.cc/2022/Conference — NeurIPS 2022 Accept_

### Official Review · Reviewer_tYtV · 2022-06-27

**Rating:** 7
**Confidence:** 4
**Soundness:** 4 excellent
**Presentation:** 3 good
**Contribution:** 3 good

**Summary:**

This work proposes a non linear MCMC method based on an interacting particle system. The article studies its long-time and large-particle non-asymptotic behaviour, ultimately showing the method consistency under standard hypotheses. The method is then illustrated on toy and real-world examples. While the paper is strongly related to [1], it the way the empirical non-linearity is formed (and subsequent analysis) differs from [1] in the following way: it is defined as a self-interacting process in [1], and as an interacting particle system in this work.

**Questions:**

1. The paper argues that MCMC consists in running independent chains and averaging the results out (line 43). However, this is typically argued against in the literature (see http://users.stat.umn.edu/~geyer/mcmc/one.html for a summary). This is also not the approach taken by [1] as the main related work. I believe why the paper considers this framework in particular probably requires some discussion. This seems a subtle yet key difference with [1], visibly coming from the IPS perspective in this work versus the self-interacting perspective of [1]. Perhaps a background section on [1] in the supplementary would help the reader pinpoint the exact difference with [1].

2. A related question is concerned with the ergodicity of the system (3): can we hope for a LLN (similar to Theorem 6.1 in [1]) for a fixed number of particles $N$?

3. Corollary 1 seems to be a corollary of Theorem 3, not Theorem 1. It would be good to refer to the proof Proposition 2.2 in [41] to help the reader. Furthermore, a square seems to be missing from the corollary as compared to the proof (although Cauchy-Schwarz gives you the result stated). If however, the desired result is indeed in terms of absolute error, why not directly use Theorem 3 with $q = N$ on the function $\tilde{f}(x_1, \ldots, x_N) := \frac{1}{N osc(f)}\sum_{k=1}^N f(x_k)$? I think this would be simpler to understand than the argument given.

4. Theorem 3: I believe the hypothesis on $J$ should be for all probability measures $\eta$. The proof is difficult to read, is it possible to make the absolute value bars larger to see where it is split? There is a typo in the proof second line of the equation block under line 888: the subcripts for the compensating terms are wrong. Furthermore, a lot of things are happening between step 3 and 4 of this block. Isn't there a term missing similar to what is obtained in Case 2? More details on the derivation would be appreciated. As minor details, there are some double closed (and unclosed) brackets and parentheses throughout the proof. E.g. line 896 and 897 (last equation of the block). There is a typo on the fact that K is a q-product on line 894. Also, the notation is not coherent in terms of applying a kernel to a function: sometimes it is K(f), sometimes Kf (and similarly for J(f) vs Jf). Can you make it fit with Section 1.3?

5. This is minor, but in Assumptions D.2., isn't the condition that V needs to be finite at least somewhere missing?

6. In the experiments section, why is the current work not compared with [1]? While I don't mind the method not necessarily reaching state of the art, I believe it is rather important to compare it to its closest relative. My guess is that this is due to a technical limitation related to using JAX. This is clearly an issue with [1] that it requires evergrowing memory (and it could be pointed in the paper) however, it would be great to compare the two methods on at least the toy examples.

7. As mentioned in the CIFAR example, using an based sampler for Q results in a biased method of X, because G is now not perfectly related to Q. In this case, have the authors considered using an empirical version of G: $G_n$, where $\eta^*$ is replaced by using the previous estimate $\eta_n$, giving (in the Boltzmann case) $K_{\eta, n} = (1-\epsilon) K(x, dy) + \epsilon \Psi_{G_n}(\eta)(dy)$ (this would be, I guess, a bit similar to [50])?

8. The run time for the different experiments was not reported. In fact, I believe that Figure 2 should be reported in terms of iterations (as here) as well as with the x-axis being the run time.

**Limitations:**

Some limitations of the work are a bit too quickly brushed over, in particular, the fact that the LLN of this article is across the particles, and not on the trajectory of the Markov chains, is a bit too subtle (see questions).

**Strengths And Weaknesses:**

# Strengths
This is clearly a strong methodological paper, with good theoretical backing for the consistency of the method. The method is creative and substantially different from previous works. The presentation is rather clear, but can be improved upon as hinted in the questions section.

# Weaknesses
The differences with [1] are fairly subtle and may not be noticed at first read, in particular because no background on [1] is provided.
This paper is rather long for a conference format, and most of the story, and interesting proof points lie in the appendix.
The experiments lack a comparison with [1] which is the main related work, rather than MALA.


# Minor comments and typos

1. The references are not formatted properly (for instance lacking proper capitalisation of names "hamiltonian monte carlo", "bayes" etc), or sometimes not properly referenced ("How good is the bayes posterior in deep neural networks really?" is published at ICML, "Uniform long-time [...]" at Journal of Statistical Physics)
2. intractible -> intractable
3. has an added -> presents an additional
4. Compatability -> Compatibility
5. critera -> criteria
6. “distribution flow” is not linked to equation (8)
7. This is inequality -> This inequality (line 222)
8. The weighted supremum norm term used in G2 is not defined explicitly in D.1.2
9. of auxiliary density $\eta^*$ is required -> the auxiliary
10. dyanmics -> dynamics
11. stepszie -> stepsize
12. otherewise -> otherwise
13. account of our experimental settings -> missing full stop.
14. which is depends -> which depends
15. Capitalisation of names in the references is sometimes off (langevin, gibbs, etc)
16. I can see that some cited papers are given as arxiv references, but are actually published (at least [32], [40], [60]), please check.

---

> ### Author Response · Authors · 2022-08-01
> **Rebuttal for Reviewer tYtV**
>
> Dear Reviewer tYtV,
>
> We offer our sincerest thanks for your incredibly detailed and comprehensive review of our work. Your many helpful questions; grammatical, spelling, and notational corrections; and careful reading of our proofs have undoubtedly led to a significantly improved draft, which we have uploaded.
>
> We wish to address some of your questions and comments below.
>
> *Re: "The differences with [1] are fairly subtle and may not be noticed at first read, in particular because no background on [1] is provided".* We originally had a section describing the differences in detail but it was moved to Appendix A.1 due to space constraints. However, during the editing process, we accidentally omitted a pointer from the main text to this appendix -- this oversight has been rectified in line 99 of the revised draft.
>
> *Re: "The experiments lack a comparison with [1] which is the main related work, rather than MALA".* We felt that, given the limited space available and the relative obscurity of these nonlinear methods, using well-known baselines such as MALA and ULA would provide the most informative examples to the widest possible audience. We 100% agree that additional experiments to compare our algorithm to its close relatives in [1] "as is" would be a valuable addition to a future version of the paper. However, we also note that running the algorithm from [1] at scale (e.g., on CIFAR10) would prove challenging due to its computational inefficiency.
>
> *Re: Questions.*
> 1. The desired background section is in Appendix A.1, but we accidentally failed to point to it from the main text after moving it there.
>
> 2. We think the answer is likely "no". Consider the BG interaction; it is easy to find cases where $\pi$ is not $\Psi_G$-invariant for any $N < \infty$ when using $m(\overline{Y}_\infty)$ instead of $\eta^\star$ (it may be ergodic w.r.t. some measure, but it won't be $\pi$). While this intuition is captured from above in Theorem 1 (i.e. the upper bound is not zero for any finite $N$), it would be  interesting to find a lower bound on total variation distance for finite $N$ for a general class of examples to strengthen this argument.
>
> 3. It's true that Corollary 1 is a direct consequence of Theorem 3, however Theorem 1 (which contains Theorem 3) is stated in the main text, so we referenced the latter instead. We should have pointed to Prop 2.2 in [41] in the proof of the corollary as it uses their arguments and is only present for completeness -- this was in a previous draft and was accidentally removed, and we have fixed this in our revised draft. We think the proof should be correct as written: convergence in law (what we are trying to show) is a direct consequence of L2 convergence (what the proof uses).
>
> 4. Yes, the condition on $J_\eta$ is a global "Lipschitz"-like condition and should hold for all $\eta$. We have made this explicit in the statement of the result now. We have also addressed the other typesetting issues, including normalizing the notation and enlarging the absolute value bars to make the proof more readable. We think that the proof for Case 1 of Theorem 3 should be correct; there is no additional compensating term for $J_\eta$ because $J_\eta f(x)$ is actually a constant function in Case 1 since $J_\eta(x, dy)$ doesn't depend on $x$ (e.g. when $J_\eta(x,dy) = \Psi_G(\eta)(dy)$). Therefore we don't need to compensate for the fact that $J_{\eta}$ is integrated with different measures as we do for Case 2 in the decomposition after line 895 -- this allows us to get a stronger result in Case 1 than Case 2. The notation might be a bit misleading in Case 1, but it is consistent with the statement of the theorem and we feel that introducing even more notation would likely not increase readability.
>
> 5. In Appendix D.2, the ranges of $V$ and $Q$ do not include infinity, so we don't think specifying that they are finite somewhere is necessary.
>
> 6. Please see above for our reasoning for using MALA as the baseline rather than the sampler in [1].
>
> 7.  This is an interesting suggestion, as this mismatch is a drawback of the methods. One potential issue, if we have understood your suggestion correctly, is that we would need to evaluate the density of $\eta_n$ in the $G_n$ proposed, but since we're using an empirical estimate this would be difficult. Despite this, your suggestion reveals an interesting connection to annealed importance sampling and needs further investigation.
>
> 8. We have added the missing runtime analysis in our revised draft, Appendices C.1.3 and C.2.4, which measure the performance vs gradient evaluations and also the wall time speed vs the linear method.

---

> > ### Comment · Reviewer_tYtV · 2022-08-04
> > **Acknowledgement of response**
> >
> > Thank you for your response. I have read other reviews (and responses thereof) and I do not change my assessment. I still believe this is a strong paper and that a comparison to [1] should have been included (if anything in the supplementary material) for completeness, but this work can stand without it.

---

> > > ### Author Response · Authors · 2022-08-09
> > > **Response to reviewer**
> > >
> > > Dear Reviewer tYtV,
> > > Thank you for your response. We plan to include the comparison to [1] in a future version of the paper.

---

### Official Review · Reviewer_RMta · 2022-07-11

**Rating:** 5
**Confidence:** 3
**Soundness:** 3 good
**Presentation:** 3 good
**Contribution:** 2 fair

**Summary:**

The work explores the existing technique of nonlinear MCMC, where the transition kernel draws information from across a collection of samples. It establishes some theoretical results, and proposes two such kernels (Boltzmann-Gibbs interaction and Accept-Reject interaction), with empirical demonstrations of both.

**Questions:**

* How are the x-axes on the MMD plot in Figure 1 scaled? Do they account for different number of kernel executions in each MCMC step for these methods (if I understand correctly), i.e. computational complexity?

**Limitations:**

No concerns.

**Strengths And Weaknesses:**

This work reflects substantial effort, but a good amount of the material reads as a review, and I'm unsure that the theoretical results really connect with the proposed kernels to make for a coherent contribution at this stage. This does look to be a useful line of inquiry and I do not wish to discourage the author(s), but the work as it stands does not yet seem ready for publication.

In a sense it seems like two works, the first establishing the theoretical results, the second proposing the new kernels within the nonlinear MCMC framework and showing they can work, although not that they can outperform. On the first I'm not well placed to determine whether the theoretical results alone stand as a sufficient contribution or display original insight (and if they do, they might be published in isolation). On the second I am not sure that there is sufficient novelty here: the kernels are not greatly different from existing work from what I can tell, and the empirical assessment of them is quite limited, showing that they work, but not that they outperform in predictive performance, say, or exhibit some other desirable quality such as accurate uncertainty quantification (as motivated in the introduction).

Overall, I think that either the method and application would need to be developed further, or the work should focus on the theoretical results, but it is difficult to recommend in its current form.

---

> ### Author Response · Authors · 2022-08-01
> **Rebuttal for Reviewer RMta**
>
> Dear Reviewer RMta,
>
> Thank you for your review and words of encouragement. We wish to address some of your comments below.
>
> *Re: "a good amount of the material reads as a review".* The reason why we devoted a considerable amount of space to a careful introduction to nonlinear MCMC methods is that these methods are not yet widely used in MCMC and likely not familiar to our target audience of ML researchers interested in Bayesian ML and sampling methods. Since we do not have the luxury of referring to many established works in the ML literature for background, we felt it necessary to work through core concepts, such as propagation of chaos or mean field vs interacting particle systems, before proceeding to the novel parts of the paper which are highly technical in nature. We think this has led to a reasonably understandable and self-contained presentation.
>
> *Re: "it seems like two works".* We acknowledge that the scope of our paper, to provide both solid theoretical analysis and realistic empirical res, is ambitious. However, we feel that reducing the scope to either one of these questions would result in an incomplete presentation. Focusing on the theory would leave open the important question of whether the proposed methods work in practice, which would underserve our target audience of ML researchers. Conversely, focusing too much on the empirical side without sufficient motivation would leave significant questions about how to proceed in future directions and the viability of these methods at all. Hence, we think that answering the closely linked questions of theory and practice in the same place is a significant strength of our contribution, and provides a strong footing from which to continue research in this direction.
>
> Also, we believe that the goals and scope of our paper are in-line with past accepted and high-impact papers to NeurIPS. Two examples are [1] and [2] (citations below), both of which explore advanced sampling methods, require significant mathematical background and setup, and provide strong theoretical results and some empirical evidence. Furthermore, we feel that our paper's empirical results more accurately reflect the type of large-scale problems that an ML researcher would encounter in their work, and that this adds additional value.
>
> *Re: "not sure that there is sufficient novelty here".*  We view our work as building on the ideas set forth in [1] (citation in our paper) and making them practical for ML, both of which contain novel contributions. More specifically, we obtain new and powerful theoretical tools for analyzing nonlinear MCMC methods, and a much more comprehensive empirical evaluation of these methods than anything in the literature we are aware of. We think that these contributions will be valuable to the Bayesian ML and probabilistic sampling communities.
>
> *Re: "the empirical assessment of them is quite limited".* While our results do not necessarily show the performance improvements we desire in high dimension, we think that they are more realistic than simple toy problems. We think that using nontrivial experimental settings will be useful to the ML community when assessing the practical efficacy of our methods. Also, we do show a small, albeit statistically significant, improvement in the calibration (a proxy for uncertainty) of the CIFAR10 models that are sampled with our methods in Table 1. Moreover, the toy problems that we study show strong improvements, so the challenge for future work will be to transfer that performance to higher-dimensional settings.
>
> *Re: Limitations.* The MMD plots' x-axes are in number of "global execution steps" $n$ consistent with Theorem 1. In the revised version of our submission, we have included quantitative runtime results (wall clock and gradient computations), please see Appendices C.1.3 and C.2.4 in the revised version.
>
> [1] Michael Arbel, Anna Korba, Adil Salim, and Arthur Gretton. Maximum mean discrepancy gradient flow. Advances in Neural Information Processing Systems, 32, 2019.
>
> [2] Qiang Liu, and Dilin Wang. Stein variational gradient descent: A general purpose bayesian inference algorithm. Advances in neural information processing systems, 29, 2016.

---

> > ### Comment · Reviewer_RMta · 2022-08-08
> > **Thanks to the authors for their response**
> >
> > I've read the authors' response and the complete set of reviews.
> >
> > Other reviewers are identifying insight and novelty in this work, especially in the theoretical results, on which I'm not well calibrated. I don't see any red flags myself, so will update my recommendation to accept. I like the direction, and it has plenty of discussion value for NeurIPS.

---

> > > ### Author Response · Authors · 2022-08-09
> > > **Response to reviewer**
> > >
> > > Dear Reviewer RMta,
> > > Thank you for your response and your time and effort in reviewing our paper.

---

### Official Review · Reviewer_3nmg · 2022-07-12

**Rating:** 7
**Confidence:** 1
**Soundness:** 4 excellent
**Presentation:** 4 excellent
**Contribution:** 3 good

**Summary:**

 This paper builds on the nonlinear MCMC methods proposed in [1], and provides convergence guarantees of these nonlinear sampling techniques in terms of the number of iterations and the number of samples required.
- Using these general convergence guarantees, the paper shows that this holds for two specific algorithms: Boltzmann-Gibbs interaction, and Accept-Reject Interaction.
- The paper also reports some positive and "negative" results on 2D toy tasks, and on CIFAR10. On 2D tasks, the paper shows that if the invariant density $\eta^*$ can incorporate further details about the problem, then nonlinear methods work really well. However, on CIFAR10 they perform as well as linear methods, but not better.

[1] Christophe Andrieu, Ajay Jasra, Arnaud Doucet, and Pierre Del Moral. On nonlinear Markov
chain Monte Carlo. Bernoulli,  17(3):987 – 1014,  2011.   doi:  10.3150/10-BEJ307.   URL
https://doi.org/10.3150/10-BEJ307

**Questions:**

1.I'm quite curious to understand and decouple the interactions between linear vs nonlinear MCMC methods, and the approximations and minibatching required to sample in a high-dimensional space such as CIFAR10. I think it might be important to decouple these interactions, to see if it really is the nature of the invariant density $\eta$ that results in similar performance to a linear MCMC scheme, or if it is one of the approximations and assumptions made in 4.2.1. Maybe the authors could consider a smaller subset of CIFAR10, or MNIST and perform full batch sampling to see if there are significant changes.

Otherwise, I am quite happy with the quality and contributions of the paper. However, as a layperson in the field of nonlinear sampling, I might have missed something important in the proofs and theorems, and would defer to the judgement of more knowledgeable reviewers before making a final judgement.

**Limitations:**

I believe the authors should refer to the additional computational requirements required by their method in terms of wall clock time as well, if possible. They might have this information in an appendix, or I might have missed it, in which case please point me to it. They have addressed social impacts.

**Strengths And Weaknesses:**

1. Originality

The paper builds heavily on the initial mathematical machinery proposed in [1]. However, they derive novel convergence bounds for these nonlinear MCMC samplers in quite general settings. They also show that these bounds are satisfied for two common nonlinear samplers. The paper also uses really interesting results from the "propagation of chaos" to show that even though the central limit theorem breaks down in a nonlinear setting due to statistical dependences across particles, there are still arguments that can be made about the independence of these interactions as $N \rightarrow \infty$. I found the motivations and mathematical proofs quite novel.

2. Quality

For someone who is a layperson to nonlinear MCMC methods, I found the paper had very high pedagogical quality, and I was able to understand to a large extent the breadth of the literature on this field. I found the mathematical proofs correct in my understanding, though I did not look closely at the proofs in the appendix. I followed through the intuition and justifications that lead to Theorem 1, and I think it is an elegant result, that gives us useful convergence bounds for nonlinear MCMC methods.

3. Clarity

I found the paper easy to follow in terms of its writing, if not always in the math. Some of this is due to a lack of expertise in nonlinear MCMC techniques, however I believe the authors did a good job providing background where necessary and shifted a large part of the verbose Mathematics to the appendices.

4. Significance

I believe this paper has quite a significant convergence bound for nonlinear MCMC. Also, I believe the empirical "negative" result on the CIFAR10 problem shows an important limitation of nonlinear MCMC methods currently. I think this result, showing that nonlinear methods perform as well as linear methods, but don't necessarily outperform them, is an important factor to consider before using these methods for large scale Bayesian machine learning.

---

> ### Author Response · Authors · 2022-08-01
> **Rebuttal for Reviewer 3nmg**
>
> Dear Reviewer 3nmg,
>
> Thank you for your helpful comments and questions, they are much appreciated. We particularly appreciate your highlight of the pedagogical quality of our work as this was something we tried hard to optimize during writing.
>
> We agree with your overall assessment of the empirical performance; the methods are interesting but suffer from some limitations as they stand now. Our paper lays a foundation of theory and baseline empirical results, and we think that deeper investigations into both the deficiencies of the current samplers in high dimension and sampler design strategies are exciting directions for future work. Your suggestion to disentangle the effects of the minibatching and other approximations fits nicely in this programme; we would like to note that we did an early experiment with MNIST (not included due to space and time constraints) at minibatch size 1000 (i.e. $\sim 4\times$ the CIFAR10 experiments) and observed similar behaviour to the CIFAR10 experiments. Other interesting considerations include understanding the impact of the number of samples (due to the particle interaction, the number of samples affects the quality of the Markov chain trajectories) and obtaining more informative choices of auxiliary target density $\eta^\star$.
>
> *Re: discussion of computational requirements.* We briefly discussed the memory requirements (which are nontrivial for MCMC algorithms on large NNs) in Section 5, lines 335-337 but not the wall clock time specifically. We also have a discussion of efficient software implementations of our algorithms in Appendix A.2, in particular how to leverage the JAX library with sample-wise and batch-wise vectorization to achieve efficient implementations. We have added a revision that has new sections in the appendix (Appendices C.1.3 and C.2.4) that report the performance vs runtime of our algorithms in wall time and number of gradient computations. These new data support the discussion in appendix A.2 on efficient implementations.

---

> > ### Comment · Reviewer_3nmg · 2022-08-09
> > **Response to rebuttal**
> >
> > I thank the authors for the reply! The authors addressed some minor concerns I had with the paper, and I believe it's a strong paper that stands to be accepted! I stand by my initial review and rating, and thank the authors once again for their comments.

---

> > > ### Author Response · Authors · 2022-08-10
> > > **Response to reviewer**
> > >
> > > Dear Reviewer 3nmg,
> > > Thank you for your response and your time spent reviewing our paper.

---

### Official Review · Reviewer_A9qb · 2022-07-25

**Rating:** 7
**Confidence:** 4
**Soundness:** 3 good
**Presentation:** 2 fair
**Contribution:** 3 good

**Summary:**

This paper considers the application of the nonlinear Markov Chain Monte Carlo (MCMC) method to problems in Bayesian inference. The key idea of the method is, instead of using a single kernel, a mixture of two kernels, a linear and non-linear one are used. Two types of non-linear kernels are considered, the Boltzmann-Gibbs interaction and the Accept-Reject interaction. It is shown that the performance of the proposed methods is at least as good as when the non-linear kernel is not used in addition to the linear one.


**Questions:**

In line 145-146 you say "jump to a new state distributed according to \eta(dy)". However, this sampler [Accept-Reject interaction] just samples from \pi using the proposal \eta^*. What is, and what is the role of \eta here?

Another question about the accept-reject interaction is why do you call it "adaptive Metropolis-Hastings"? This term is usually used for MCMC samplers that progressively change the proposal as the chain is run, while preserving convergence properties. The Accept-Reject interaction as presented here is a standard independence (or independent) Metropolis-Hastings sampler with the proposal distribution \eta^*.

I am not sure that the non-reversibility of BG (as opposed to AR) matters so much in the neural networks experiment. Have you tried AR for completeness? Does it do the same? I would presume that using a tempered density as the proposal is the more important factor here?


**Limitations:**

Yes.

**Strengths And Weaknesses:**

In general, combining different MCMC methods to achieve better exploration of a target density is good practice, e.g., in the case of a multimodal density, one can combine local moves that explore within a mode with moves that explore between different modes. The rationale is that neither of these moves by themselves would provide good exploration of the target density: local moves will take a long time to travel between different modes, whereas global moves will typically have trouble exploring a mode locally. This paper aligns with the general idea of the benefit of combining together different MCMC kernels.

The general idea of the paper is sound. However, a drawback is that it is at times hard to understand what exactly is happening and to link various portions of the paper together. As far as was able to tell, the idea at the core of the paper is as described above, to combine MCMC moves with different average jump sizes and to show this leads to better exploration. For example, the description of an Interacting Particle System is hard to link to the concrete MCMC schemes (i.e., based on the Boltzmann-Gibbs and Accept-Reject interaction) proposed in this paper.

---

> ### Author Response · Authors · 2022-08-01
> **Rebuttal for Reviewer A9qb**
>
> Dear Reviewer A9qb,
>
> Thank you very much for your review and helpful questions. We wish to address some of your comments below.
>
> *Re: the link of the IPS to the MCMC schemes.* Section 2.2, lines 113-128 in the original submission attempt to do precisely this; essentially, the MCMC schemes are instances of the IPS for particular choices of $K$, $J_\eta$, and $Q$ that ensure $\pi$-invariance, etc. We felt it was necessary to spend considerable space on the general setup and then specialize to MCMC because nonlinear and jump MCMC are far less common in the literature, and the background is essential for understanding our analysis in Section 3.
>
> *Re: "this sampler [Accept-Reject interaction] just samples from $\pi$ using the proposal $\eta^\star$. What is, and what is the role of $\eta$ here?"*. We think there may be a slight point of confusion here; the A-R sampler actually uses the empirical law of the Markov chain $\eta_n$ as the proposal at step $n$, not $\eta^\star$, and computes the acceptance ratio with $\eta^\star$. This balances the fact that sampling from $\eta^\star$ (ideal case) may not be feasible, and knowing the density of $\eta_n$ (to compute the acceptance ratio) is generally impossible. As $\eta_n\to \eta^\star$, the limit is indeed sampling from the proposal $\eta^\star$.
>
> *Re: "why do you call it "adaptive Metropolis-Hastings"?".* We think this is answered in the previous comment; in particular, the proposal distribution evolves as the empirical law of the auxiliary Markov chain as we noted in lines 148-149.
>
> *Re: Importance of BG non-reversibility & trying AR.* We did try the AR sampler for the neural network experiments and observed that the mixing was incredibly slow due to very low acceptance ratios after a couple of steps. We observe the same behaviour with MALA vs ULA. We acknowledge that it would have been better to include these "negative" results in an appendix and plan to do so in a future revision.

---

> > ### Comment · Reviewer_A9qb · 2022-08-08
> > **Thanks for clarifying**
> >
> > Thank you for clarifying, I have adjusted my review score. I am still not entirely sure why the AR would be much slower, so inclusion of these results would be appreciated.

---

> > > ### Author Response · Authors · 2022-08-09
> > > **Response to reviewer**
> > >
> > > Dear Reviewer A9qb,
> > > Thank you for your response. We plan to add the AR results in a future version.

---

### Meta-Review · Area_Chair_dK6e · 2022-08-26

**Recommendation:** Accept
**Confidence:** Certain

**Metareview:**

The contribution of this submission is strong - an analysis of convergence of non-linear MCMC methods. Most reviewers agree that the submission is theoretically interesting and of interest to the NeurIPS community. Thus I recommend acceptance.

However, I note (and share with some reviewers) the following concern: the empirical results on CIFAR10 do not show the benefit of the proposed method. The result of the linear baseline (ULA) is very far from SOTA for Resnet and Cifar10, yet ULA outperforms the non-linear version in terms of accuracy and time efficiency. It would be great if there is a realistic model/dataset (between the simple 2D toy experiment and perhaps too ambitious resnet/cifar10) that can be included to show the benefits (if any) of nonlinear MCMC. Some reviewers also raised a concern about the organisation/flow of the paper which I hope the authors will fix in the camera-ready version.

**Award:**

No

---

### Decision · Program_Chairs · 2022-09-14

Accept